# Poisonous Piperidine Plants and the Biodiversity of Norditerpenoid Alkaloids for Leads in Drug Discovery: Experimental Aspects

**DOI:** 10.3390/ijms232012128

**Published:** 2022-10-12

**Authors:** Ashraf M. A. Qasem, Michael G. Rowan, Ian S. Blagbrough

**Affiliations:** School of Pharmacy, University of Bath, Bath BA2 7AY, UK

**Keywords:** *Aconitum*, biodiversity, *Delphinium*, drug discovery, norditerpenoid alkaloids, taxonomy

## Abstract

There are famous examples of simple (e.g., hemlock, *Conium maculatum* L.) and complex (e.g., opium poppy, *Papaver somniferum* L., Papaveraceae) piperidine-alkaloid-containing plants. Many of these are highly poisonous, whilst pepper is well-known gastronomically, and several substituted piperidine alkaloids are therapeutically beneficial as a function of dose and mode of action. This review covers the taxonomy of the genera *Aconitum*, *Delphinium*, and the controversial *Consolida*. As part of studying the biodiversity of norditerpenoid alkaloids (NDAS), the majority of which possess an *N*-ethyl group, we also quantified the fragment occurrence count in the SciFinder database for NDA skeletons. The wide range of NDA biodiversity is also captured in a review of over 100 recently reported isolated alkaloids. Ring A substitution at position 1 is important to determine the NDA skeleton conformation. In this overview of naturally occurring highly oxygenated NDAs from traditional *Aconitum* and *Delphinium* plants, consideration is given to functional effect and to real functional evidence. Their high potential biological activity makes them useful candidate molecules for further investigation as lead compounds in the development of selective drugs.

## 1. Piperidines: Poisonous, Tasty, and Beneficial

Nature is rich with examples of plants that can be described as medicinal or poisonous. The controversy in the description spotlights the fact, well known in pharmacy and the pharmaceutical sciences, that dose and mode of action are critical. Out of thousands of plants with various types of active principles, many examples can be found of natural sources of alkaloids containing a substituted piperidine nucleus.

There are well-known examples of simple piperidine-alkaloid-containing plants, such as the famous poison hemlock (*Conium maculatum* L.), in the family Apiaceae (formerly Umbelliferae). It is a lethal poison that was given to criminals in ancient Greece and that the Greek philosopher, Socrates, was forced to drink (399 B.C.) [1]. The principal component of poison hemlock is the piperidine alkaloid coniine **1** (Figure 1), which is a nicotinic acetylcholine receptor (nAChR) agonist [1] where the importance of the positive centre was highlighted in the Beers–Reich model [2]. Consumption of *C. maculatum* leads to various degrees of toxicity in animals, where it has been found that it is more poisonous to cattle than to other animals. Human toxicity signs were described by Socrates’ pupil as trembling, staggering, and rapid muscular weakness. Death resulting from hemlock poisoning is mainly due to respiratory failure [1,3]. Another nAChR agonist is anabasine **2**, a natural nicotine-**3**-like isomer compound from *Nicotiana* spp. (tobacco) of the family Solanaceae. Two important species are *N. glauca* (wild tree tobacco) and *N. tabacum* L. Anabasine **2** is the major component in *N. glauca*, while nicotine **3** is the main constituent in *N. tabacum* L. [3,4,5]. Another source of piperidine alkaloids is *Lobelia* spp. (Campanulaceae). An important example is the Indian tobacco, *L. inflata* L., which is used traditionally in smoking cessation and in the treatment of respiratory conditions [6]. Lobeline **4** is the major and the most biologically active component of *L. inflata* L. It was found that lobeline **4** can be described as agonist, antagonist, or mixed agonist/antagonist at nAChR [7].

*Lupinus* spp. (Fabaceae) consumption leads to teratogenic crooked calf disease. The teratogenicity was initially suggested to be due to the quinolizidine alkaloid, anagyrine **5** [5,8]. As similar teratogenicity happens in livestock due to the consumption of *C. maculatum* and *N. glauca*, which contain mainly piperidine alkaloids [4,9,10], it is now suggested that the main piperidine alkaloid, ammodendrine **6**, which is found in many *Lupinus* spp. such as *L. formosus*, is responsible for such deformations. Keeler and Panter [8] showed that anagyrine **5** exists as a minor component in *L. formosus*, while ammodendrine **6** is the major one, supporting that the piperidine alkaloid **6** is responsible for the teratogenicity.

Black pepper, *Piper nigrum* L. (Piperaceae), is known as the king of spices, and it is the natural source of piperine **7** which shows anti-inflammatory, antioxidant, anticancer, and antimicrobial activities [11,12,13]. 1-Deoxynojirimycin **8** is another example of a piperidine alkaloid which acts as one of the most potent α-glycosidase inhibitors and has relevant biological activity in the treatment of hyperglycaemia and obesity. It is naturally occurring in the leaves of white mulberry, *Morus alba* L. (Moraceae) [14,15].

In addition to simple piperidines, there are many famous examples of piperidine-containing plants where the piperidine ring is part of a complex skeleton. *Papaver somniferum* L. (Papaveraceae), also known as opium poppy, is a natural source of the opioids morphine **9** and codeine **10**. These piperidine-containing alkaloids are used as strong analgesics, but are also abused in addiction [16,17]. The potency of morphine **9** is much higher than codeine **10**, and the fact that codeine **10** exerts its analgesic effect after being metabolized into morphine **9** highlights the importance of the phenolic alcohol in the activity at μ opioid receptors [18]. Another example is cytisine **11**, which is derived from *Laburnum anagyroides* Medik. (Fabaceae), and it acts as an nAChR agonist. The skeleton of cytisine **11** shows that the piperidine coexists with a quinolizidine ring [3,19] and fits the Beers–Reich model, as cytisine **11** contains a cationic centre and heteroatoms, where the model showed that the distance between them in nAChR ligands is 5.9 Å [2].

There are piperidine-containing alkaloids, which are C-18 and C-19 norditerpenoid alkaloids (NDAs) from *Aconitum* and *Delphinium* (Ranunculaceae), and especially aconitine **12**, lappaconitine **13**, lycoctonine **14**, lycaconitine **15**, and methyllycaconitine (MLA) **16**. The C-19 NDA aconitine **12** was first discovered in 1833 by P. L. Geiger from *A. napellus* [20,21]. It is considered a potent lethal cardiotoxin that acts on voltage-gated sodium channels (VGSC) and keeps them in an open conformation. In contrast, lappaconitine **13**, which was the first C-18 NDA to be discovered, is a VGSC blocker [22]. The hydrogen bromide (HBr) salt of lappaconitine **13** (allapinine) is used clinically in Russia as an anti-arrhythmic drug. Lappaconitine **13** was first discovered from *A. septentrionale* Koelle by H. V. Rosendahl in 1895 [23,24]. Lycoctonine **14** was first reported in 1865 from *A. lycoctonum* L. [25]. Lycaconitine **15** was first reported in 1884 from *A. lycoctonum* L. [26]. MLA **16**, a C-19 NDA, was first discovered by Manske in 1938 in *D. brownii* Rydb [27]. Goodson (1943) later determined its exact formula [28]. MLA **16** acts as a potent competitive antagonist on α7-nAChR [29]. The importance of the piperidine nitrogen of NDAs in the interaction with nAChR was proven, as semi-synthetic analogues with quaternary nitrogen showed higher activity, and that suggests that the nitrogen is a main site of receptor interaction [30]. All the previous examples show alkaloids which contain piperidine alone or they coexist with other heterocycles. Scopolamine (hyoscine) **17**, L-hyoscyamine **18**, and atropine (dl-hyoscyamine) **19** (Figure 1) are examples of alkaloids that contain a piperidine ring fused with pyrrolidine (tropane alkaloids). These alkaloids are antagonists at the muscarinic acetylcholine receptors (mAChR) and are derived from many plants of the nightshades (Solanaceae). These compounds meet the Beers–Reich pharmacophore criteria, as they contain a cationic centre and a Van der Waals surface (heteroatoms). A member of the nightshades is *Datura stramonium* L., which is also known as thornapple and jimsonweed [31,32].

## 2. Taxonomy of Aconitum, Delphinium, and Consolida

Diterpenoid alkaloids are found mainly in *Aconitum*, *Delphinium*, and *Consolida* within the family Ranunculaceae, and *Garrya* (silk tassel) from the family Garryaceae. Apart from these genera, three diterpenoid alkaloids, lycoctonine **14**, MLA **16**, and inuline (which is the 2-aminobenzoate ester of lycoctonine **14**) have been reported from *Inula royleana* (Asteraceae) [33]. The three families (Ranunculaceae, Garryaceae, and Asteraceae) are classified under the Angiospermae class (flowering plants) [34,35,36]. C-18 and C-19 NDAs are derived from only three genera within Ranunculaceae: *Aconitum*, *Delphinium*, and *Consolida*. The Ranunculaceae family contains around 43 genera and more than 2000 species [37]. *Aconitum* L. with around 330 species and *Delphinium* L. with around 450 species are considered the major genera in this family [38,39].

All three of these genera are classified within the tribe Delphinieae of the subfamily Ranunculoideae. [40]. A scientific classification of *Aconitum*, *Delphinium*, and *Consolida* is shown in Figure 1 [36,40,41,42].

*Aconitum* L. has been divided into three subgenera (*Aconitum*, *Lycoctonum* (DC.) Peterm., and *Gymnaconitum* (Stapf) Rapes). The *Aconitum* subgenus *Aconitum* produces biennial tuberous roots, while subgenus *Lycoctonum* (DC.) Peterm. species have perennial rhizomes. The only annual species of the *Aconitum* genus can be found in subgenus *Gymnaconitum* (Stapf) Rapes. [43]. The *Delphinium* L. genus is also divided into two subgenera (*Delphinastnim* (DC.) Wang and *Delphinium*) [44], and the species within this genus are usually perennial (occasionally annual) [45]. Due to the shape of flowers of the *Delphinium* species which resemble dolphins, the *Delphinium* genus takes its name from the Greek word *delphis* [45].

The genus *Consolida* has proved to be more controversial. A. P. De Candolle separated a group of annual species from the genus *Delphinium* L. to form an independent section (*Consolida* DC.). S.F. Gray changed the section *Consolida* to the rank of a genus in 1821 (*Consolida* (DC.) S.F. Gray). Boisser gave the rank of genus to the *Consolida* section of *Delphinium* L. in 1867 and Huth gave it the rank subgenus in 1895 [46]. Huth was the last worker to include *Consolida* within *Delphinium*. Much more recently, Jabbour and Renner (2011) suggested using a DNA phylogenetic study that *Consolida* should be embedded in *Delphinium* [47]. Commonly, *Delphinium* and *Consolida* species are called larkspur, which is also a name derived from the shape of the flowers [45]. The colourful flowers of the *Delphinium* species gave rise to many cultivated species (cultivars) that are used as ornamental plants in the garden. These hybrid species come from crossing different parent plants and mainly from the tetraploid *D. elatum* L. [48,49]. Examples of the hybrid *Delphinium* varieties are the giant pacific court hybrids which originate from *D. elatum* and other species such as *D. exaltatum* and *D. formosum* [50].

## 3. NDA Chemical Toxicity

North Americans have divided the *Delphinium* (larkspur) plants into three categories depending on habit of growth and environment. First are the tall larkspurs (such as *D. barbeyi*, *D. occidentale*), which are 1–2 m tall and generally exist at altitudes above 2400 m in moist habitats. Second are the intermediate larkspurs (such as *D. geyeri*, plains larkspur), which are 0.6–1 m tall and grow on the short grass prairies of Nebraska, Wyoming, and Colorado. The third category is low larkspurs (such as *D. andersonii*), which are less than 0.6 m tall and generally grow in the desert/semidesert, foothills, or low mountain ranges [51,52]. *Delphinium* (larkspur) alkaloids cause economically important livestock toxicity across North American ranges [51,53]. Tall larkspurs contain higher amounts of toxic NDAs and are therefore considered a greater threat [54]. Intoxication happens due to the action of NDAs at the α1-nAChR expressed at neuromuscular junctions (NMJ) [55].

It was found that the livestock intoxication by larkspurs is controlled by different factors, for example, cattle breed and genetics affect the susceptibility to the intoxication. Age is another factor, where young heifers are more susceptible than mature cows. The cattle sex was reported to be an effective factor, where heifers are more prone to the toxicity than steers and bulls. Lastly, the plant factor plays an important role, where the alkaloid concentration and composition of methylsuccinimidoanthranoyl-lycoctonine (MSAL) and non-MSAL (Figure 2), which depend on the population, species, climate, and the year, affect the toxicity in cattle and the amount needed to develop clinical signs [56,57].

The toxicity of NDAs found in three tall larkspur species (*D. barbeyi, D. occidentale, D. glaucescens*) was tested in mice. The assay revealed that the 7,8-methylenedioxy-lycoctonine (MDL) alkaloids are the least toxic NDAs. The lycoctonine-type is twice as toxic as MDL, but it is considered to be a low toxic group, where the least toxic alkaloid of this category, brownine **20**, has a toxicity which is comparable to the MDL NDA. The MSAL alkaloids MLA **16** and 14-deacetylnudicauline **21** were 10-times more toxic than any other tested NDAs (Figure 2) [54].

MSAL is much more toxic than MDL, and the MSAL level in the tall larkspurs mainly contributes to livestock poisoning. A report investigated the importance of the MDL alkaloids and found that MDL alkaloids exacerbate the toxicity of the MSAL alkaloids; as the ratio of MDL to MSAL increases, the amount of MSAL that is needed to develop clinical signs decreases. The exact mechanism of this MDL action is not known, but it was suggested that MDL may act as a co-agonist in an allosteric manner or at the orthosteric ligand binding site of the receptor to exacerbate the toxicity of MSAL-type alkaloids on nAChR and therefore increase their toxicity [58]. The observed action could also be due to an effect of MDL alkaloids on metabolic enzymes which results in prolonged exposure to the MSAL alkaloids, but further investigation is needed.

## 4. Norditerpenoid Alkaloid (NDA) Biodiversity

NDAs have complex highly oxygenated hexacyclic systems, and as many of them are of pharmacological importance, their structures and 3D configuration are significant factors in their actions at various biological targets [59]. The majority of NDAs possess an *N*-Et group, as shown in Table 1, which shows various NDA skeletons and their abundance in the SciFinder database. Substitution at position 1 is important to determine the NDA skeleton conformation, as ring A in 1-OMe NDA free bases exists in a twisted-chair conformation, and in 1-OH NDA ring A adopts a twisted-boat conformation [60,61]. Table 1 shows that 1-OMe NDA abundancy is 10-times higher than 1-OH NDA. The biological activity of NDAs attracts natural product chemists to investigate their sources, and that has resulted in the discovery of interesting NDA skeletons, some of them with pharmacological importance.

Recent phytochemical investigations reported on *Aconitum* and *Delphinium* species show the wide variety of structural motifs in such NDAs. Chen et al. reported the extraction of a new aconitine-type NDA with a 1-OH substitution, pubescensine **22** from *A. soongaricum* var. pubescens, and this showed a potent insect antifeedant activity (EC_50_ < 1 mg/cm^2^) [62]. Ding and co-workers discovered four new NDAs, vilmorines A–D **23**–**26** from *A. vilmorinianum* (Figure 3) [63]. Vilmorine D **26** exhibited moderate to weak antioxidant activity (Fe^2+^ chelation activity) with IC_50_ = 33.6 ± 0.2 μg/mL, and it showed antibacterial activity against *Staphylococcus aureus* and *Bacillus subtilis* with MICs of 64 and 32 μg/mL, respectively. Vilmorine A **23** has an unusual spiro junction. Only three such compounds were isolated with that characteristic skeleton (Table 1). Vilmorine A **23** also has the really unusual 1-β-OMe group, whereas the vast majority of the position 1 substituents have an α-configuration. Vilmorines B–C **24**–**25** have an unusual cyclopropyl moiety, and they are rare examples containing an imine (piperideine) (Table 1).

Qin et al. isolated five new NDAs from *A. carmichaelii*, carmichaenine A–E **27**–**31** [64], with characteristic 1-OH substitutions. Majusine D **32** from *D. majus* W. T. Wang and stapfianine A **33** from *A. stapfianu* were discovered as new C-19 NDAs [65,66] with 1-OH substitutions. Sharwuphinine B **34** was discovered from *D. shawurense* as one of the few quaternary C-19 NDAs (Figure 4) [67].

Chen et al. isolated two imine (piperideine)-type NDAs, vilmorrianines F–G **35**–**36**, in addition to new *N*-desethyl-*N*-formyl-8-*O*-methyltalatisamine **37** from *A. vilmorinianum* Komarov [68]. Six new NDAs, 6-dehydroeladine **38**, elapacidine **39**, iminopaciline **40**, iminoisodelpheline **41**, iminodelpheline **42** (piperideines), and *N*-formyl-4,19-secopacinine **43**, were extracted from *D. elatum* seeds (Figure 4) [69]. Shan et al. reported two new C-18 NDAs, anthriscifoltine A–B **44**–**45** from *D. anthriscifolium* var. majus [70]. Ding and co-workers also isolated three new NDAs, vilmotenitines A–C **46**–**48** from *A. vilmorinianum* var. *patentipilum*, (Figure 5) where vilmotenitines A and B **46**–**47** had an unusual (spiro) rearranged six-membered B ring [71], as they had found in vilmorine A **23** with the inverted substituent stereochemistry at position 1 [63].

Two new C19 NDAs, iliensine A and B **49**–**50**, were isolated from *D. iliense*, where iliensine A **49** had a characteristic glycosidic linkage [72]. Wada et al. isolated four new C19 NDAs with a 7,8-methylenedioxy moiety from *D. elatum* [73], 19-oxoisodelpheline **51**, *N*-deethyl-19-oxoisodelpheline **52**, *N*-deethyl-19-oxodelpheline **53**, and melpheline **54** (Figure 6).

Wang and co-workers discovered three new C19 NDAs, szechenyianine A, B, and C **55**–**57**, from *A. szechenyianum* [74]. All three compounds were tested against nitric oxide (NO) release inhibition, as they were considered potential anti-inflammatory agents. Szechenyianine A **55** showed activity with IC_50_ 36.6 ± 7 μM, while szechenyianine B **56** had IC_50_ 3.3 ± 0.1 μM, and that highlights the importance of the N-O moiety. Szechenyianine C **57** (Figure 6) which is a 7,17-secoaconitine-type NDA, also showed potent activity, with IC_50_ 7.5 ± 0.9 μM.

Chao Zhan et al. reported caerudelphinine A **58**, a new 1-OH C-19 lycoctonine-type NDA from *D. caeruleum* Jacq. ex Camp [75]. Grandiflorine B **59** was also reported as a new C-19 lycoctonine-type NDA from *D. grandiflorum* [76]. The unusual skeleton of grandiflorine B **59** shows cleavage of the 7–17 bond and *N*-C19 bond and the formation of an unusual *N*-C7 bond. Zhao et al. reported three new NDAs, nagaconitine A–C **60**–**62** from *A. nagarum* var. heterotrichum [77]. Nagaconitine A **60** has a unique acyl group which was reported in only three NDAs. 8,14-Diacetate diester **62** showed antitumor activity against cancer cell line SK-OV-3. Two more new C-19 NDAs, 14-benzoylliljestrandisine **63** and 14-anisoylliljestrandisine **64**, were isolated from *A. tsaii* (Figure 7) [78].

Extensive phytochemical studies on *D. anthriscifolium* var. majus led to the isolation of six new C-19 NDAs with the 7,8-methylenedioxy moiety, which were named anthrisciflorine A–F **65**–**70** (Figure 8) [79].

Guo et al. isolated two new NDAs, 7,8-epoxy-franchetine **71** and N-(19)-en-austroconitine **72**, from *A. iochanicum* [80]. Tested against NO production in macrophages (mouse cell line), they showed a weak anti-inflammatory effect. Liang et al. isolated sinchiangensine A **73** as a new NDA from *A. sinchiangense* W.T. Wang (Figure 9), and it showed significant antitumour activity against cancer cell lines A-549, SMCC-7721, MCF-7, and SW-480 [81]. The IC_50_ (μM) values of **73** against these cell lines were 12.8, 9.6, 11.8, and 18.8, respectively, and these values were comparable with cisplatin, the positive control, the IC_50_ (μM) values of which were 22.3, 18.6, 28.8, and 18.2. Sinchiangensine A **73** also showed potent antibacterial activity against Gram-positive *S. aureus* ATCC-25923, with an MIC value (μmol/mL) of 0.15 which is comparable to 0.67, the MIC of the positive control berberine HCl.

Ajacisines A–E **74**–**78** (Figure 10) were isolated as new NDAs from *D. ajacis* [82]. Testing the in vitro antiviral activity against respiratory syncytial virus (RSV), compounds **76**–**78** showed moderate to weak effects. The IC_50_ (μM) values of **76**–**78** against RSV were 75.2 ± 1.1, 35.1 ± 0.6, and 10.1 ± 0.3, respectively, while the IC_50_ (μM) of ribavirin (the positive control) was 3.1 ± 0.8 [82].

Meng et al. reported the isolation of four new C-19 NDAs, aconicarmichoside A–D **79**–**82** (Figure 11), from the aqueous extract of fuzi and the lateral roots of *A. carmichaeli* [83]. These four alkaloids are the first examples of glycosidic NDAs where the glycosides are directly attached to the alkaloid skeleton.

Fukuyama and co-workers [84] achieved the synthesis of cardiopetaline **83** through Wagner–Meerwein rearrangement of the denudatine skeleton into an aconitine skeleton without the need of pre-activation of the hydroxy group. This means that there is no need to differentiate the hydroxy groups in the poly-oxygenated system as was needed before. Sarpong and co-workers [85] developed a unifying strategy to synthesize C-18 NDA (weisaconitine D **84**) and C-19 NDA (liljestrandinine **85**) from a common intermediate. Liu and Qin [86] highlighted the importance of dearomatization of aromatic compounds that yield *o*-benzoquinones coupled with Diels–Alder cycloaddition in the synthesis of complex structures such as **83**, **84**, **85**, and many others. Yang et al. constructed a unique tricyclo [6.2.1.0] BCD-system **86** of the NDA skeleton (Figure 12) [87].

Lian et al. reported five new C-18 NDAs, anthriscifoltines C–G **87**–**91** from *D. anthriscifolium* var. majus (Figure 12) [88]. Song et al. reported three new C-19 NDAs, szechenyianine D–F **92**–**94** from *A. szechenyianum* (Figure 13) [89].

Another four new C-19 NDAs, elapacigine **95**, *N*-deethyl-*N*-formylpaciline **96**, *N*-deethyl-*N*-formylpacinine **97**, and *N*-formyl-4,19-secoyunnadelphinine **98** (Figure 14), were isolated from *D. elatum* cv. Pacific giant [90].

Yamashita et al. also reported four new C-19 NDAs, 14-anisoyllasianine **99**, 14-anisoyl-*N*-deethylaconine **100**, *N*-deethylaljesaconitine A **101**, and *N*-deethylnevadensine **102**, from *A. japonicum* subsp. *subcuneatum* (Nakai) Kadota (Figure 15) [91].

Li et al. isolated four new C-19 NDAs, carmichasines A–D **103**–**106** from *A. carmichaelii* Debeaux (Figure 15) [92]. Extraction of the roots of *A. taronense* Fletcher et Lauener, which has been used in traditional Chinese medicine (TCM) to treat rheumatism and arthritis, yielded four new C-19 NDAs, taronenines A–D **107**–**110** (Figure 16) [93]; NDAs **107**, **108**, and **110** exhibited anti-inflammatory activity when tested in mice cells.

A study conducted on the *D. pseudoaemulans* C. Y. Yang et B. Wang resulted in the isolation of eight new C-19 NDAs, pseudophnines A−D **111**–**114**, pseudorenines A−B **115**–**116**, and pseudonidines A−B **117**–**118** (Figure 17) [94].

Ahmad and co-workers reported the isolation of two new C-19 NDAs, jadwarine A–B **119**–**120** from *D. denudatum* [95]. They also reported a new lycoctonine-type C-19 NDA, swatinine C **121** (Figure 18) which showed competitive inhibitory activity on acetylcholinesterase (AChE) and butyrylcholinesterase (BChE) [96].

Extraction of the roots of *A. brevicalcaratum* led to the isolation of three new C-19 NDAs, brochyponines A–C **122**–**124** (Figure 18) [97]. Abjalan et al. discovered a new lycoctonine-type C-19 NDA, aemulansine **125** from *D. aemulans* Navaski, which showed in vitro cytotoxicity [98]. Two novel 8,15-seco C-19 NDAs, nagarine A **126** and B **127** (Figure 19), were isolated from *A. nagarum* [99].

The variation in pharmacological activities of the NDAs, despite their structural similarities, is an attractive aspect for synthetic chemists to work on to obtain a better understanding of the structure–activity relationships (SAR). Liu et al. reported a total synthesis of the ABCDE system of the C-19 NDAs [100]. Another study reported the construction of the AEF ring system attached to a phenyl group as an analogue to ring D [101]. The construction of the fused CD-bicycle of aconitine was also achieved [102]. Lv et al. built the hexacyclic ring system of franchetine **128**, a 7,17-seco NDA [103]. The importance of such NDAs continues to attract chemists to attempt to make a total synthesis of them. Progress has been made in the total synthesis of aconitine **12**, but the construction of a pentacyclic system of the aconitine skeleton failed [104]. On the other hand, a total synthesis of talatisamine **129** (Figure 20) has been completed in 33 steps [105]. In addition, the synthesis of a [6-6-6] ABE-tricyclic analogue of MLA **16** has been achieved [106]. A synthetic approach has also been established for the BCD-tricyclic system [107].

## 5. Conclusions

It is clear that nature is rich with many examples of plants that can be described as medicinal or poisonous. The poisonous piperidine plants show that dose and mode of pharmacological action are critical. The biodiversity of natural sources of NDAs, based upon a substituted piperidine nucleus, are important for continuing to provide leads in drug discovery. NDAs from *Aconitum* and *Delphinium* have complex, highly oxygenated hexacyclic systems, and as many of them are of pharmacological importance, their structures and 3D configuration are significant factors in their actions at various protein targets with respect to medicine and toxicology.

The majority of NDAs possess an *N*-Et group. We investigated the occurrence count in the SciFinder database for NDA skeletons, including many new NDAs. Substitution at position 1 is important to determine the NDA skeleton conformation, as ring A in 1-OMe NDA free bases exists in a twisted-chair conformation. In 1-OH NDA, ring A adopts a twisted-boat conformation. In conclusion, screening NDAs for their biological activity has resulted in the discovery of new sets of ligands. These are promising natural compounds that are pharmacologically active. These hits potentially eventually will become selective leads for the treatment of a wide variety of disease states. These NDAs are new natural products, and if they can be isolated from easy-to-grow *Aconitum* or *Delphinium* plants, then the future is bright for further NDA development based on experimental aspects, including phytochemistry leading to SAR studies and hopefully to new, selective, if not specific, drugs.

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
