# Peer review of "Poisonous Piperidine Plants and the Biodiversity of Norditerpenoid Alkaloids for Leads in Drug Discovery: Experimental Aspects"

_ijms, 2022, doi:10.3390/ijms232012128_

Round 1

Reviewer 1 Report

In this manuscript, authors summarized the taxonomy of the genera Aconitum, Delphinium, and the controversial Consolida, and emphatically described the association between toxicity and biodiversity toward norditerpenoid alkaloids. Meanwhile, authors had also quantified the fragment occurrence count in the SciFinder database for Norditerpenoid Alkaloids (NDA) skeletons. Moreover, the authors found that ring A substitution at position 1 is crucial for NDA skeleton conformation. Overall, this article has certain significance for understanding NDA and provide a promising perspective for medicine development. However, despite of these interestingly results, the manuscript needs minor revisions before possible publication in Int. J. Mol. Sci. The comments are provided below.

1.     Every abbreviation should be given full name when it first appeared in abstract, text, figures and tables. For example, in the part of “Abstract”, line 11, change “Norditerpenoid Alkaloids” into “Norditerpenoid Alkaloids (NDA)”.

2.     Keywords are correct, but they are not precise enough. For example, “Norditerpenoid Alkaloids”, “Drug Discovery” “Biodiversity” should be supplemented in the keywords.

3.     In the Part of “Affiliation”, “School of Pharmacy” should not be superscripted.

4.     The Figure should be added notes under the structural formulas.

5.     Suggested edits to the text:

a.      Page 1, line 12; page 6, line 200; page 13, line 251; page 15, line 283 and page 21, line 400, “N” should be changed to “N(italic)”.

b.     Page 4, line 123, “The” should be changed as “the”.

c.      Page 15 and Page 16, disordered numbers and symbols need be revised.

d.     Line 265, change “Nagaconitine A 60 has” into “Nagaconitine A 60 had”.

5.   Line 345, “R1” should be revised as “R1(superscript)”.

6.   Uniform adjustment of author name format, e.g., references 7.

Reviewer 2 Report

Dear authors

After reviewing the manuscript, my main impression is that the paper titled “Poisonous Piperidine Plants and the Biodiversity of Norditerpenoid Alkaloids for Leads in Drug Discovery: Experimental Aspects”, is interesting but should be arranged, following the comments bellow

-The abstract:  to be more understandable and clearer, you have to manage your abstract so  

                        that the reader can understand your scientific approach and understand the  

                         general idea of your work

- The body of your manuscript contains a lot of useful information, but these data are badly arranged. You have to put your thoughts in order. It is better if you illustrate your ideas and make them more readable for the reader.

Reviewer 3 Report

Good work and hope that my comments will be helpful to improve your review.

Some specific comments are as follows:

First: The author must add a caption for Figure 1 (page 3) for those cited structures within the text and then refer to Figure 1 in their related paragraphs.   

The paragraph on page 2, Lines 56-61. The author refers to piperidine alkaloids of pepper and the leaves of white mulberry, but they are not poisonous to be mentioned under the title “1. Poisonous Piperidines”….Please delete it and their related structures from Figure 1 or change the title if he needs to mention both poisonous and non-poisonous piperidine alkaloids.

Page 2, Line 74…….between then in nAChR……. Correct to between them in nAChR

Page 2, Line 79…….voltage gated sodium channels (VGSC)….. add abbreviation in the first mentioned site.

Page 2, Line 81……. HBr salt….. add the full name for the first time before its abbreviation.

Page 2, Line 85……. Methyllycaconitine (MLA)….. add abbreviation in the first mentioned site.

Page 4, Lines 141-143, Scheme 1……Please expand the text box of (Subdivision) and (Class).

Page 5, Line 174…….N-(methylsuccinimido) anthranoyllycoctonine (MSAL) )….. add abbreviation in the first mentioned site.

Page 5, Line 179…….7,8 methylenedioxylycoctonine (MDL)……… add the full name for the first time before its abbreviation.

Structures of compounds on pages 6, 10, 11, 12,…..must have Figure numbers with related captions to be easily cited within their related text, not only structures’ numbers.

Page 13, Line 255, against NO release inhibition……. NO is it nitric oxide? Please add the full name for the first time before its abbreviation.

Page 15, Line numbers 277-282 inserted within the number of structures.

All structures need captions to be clearer for readers.

Page 15, Lines 284 and 286, A. iochanicum, A. sinchiangense………………. A. iochanicum, A. sinchiangense (Italics).

Line numbers on page 16 are inserted within the number of structures.

Page 20, Line 362… D. denudatum…………. D. denudatum (Italics)

Page 20, Line 376…… Structure-Activity Relationship (SAR). add the full name for the first time before its abbreviation.

- In the current state, there are more typographical errors. Therefore, the authors are advised to recheck the whole manuscript for improving the language and structure carefully.
